# Evaluation of Aphid Resistance and Oleoresin Production in Indigenous Tropical Pine (*Pinus merkusii* Jungh. and de Vriese)

Purwanto [1], Liliana Baskorowati [2], Pujo Sumantoro [1], Rina Laksmi Hendrati [2], Mudji Susanto [2], Mashudi [2], Dedi Setiadi [2], I.L.G. Nurtjahjaningsih [2], Sugeng Pudjiono [2], Agus Kurniawan [3], Pandu Yudha Adi Putra Wirabuana [4,*] and Sumardi [2]

1. Department Research and Innovatiom, Perhutani Forest Institute, Cepu, Blora 58302, Indonesia; purwantopusbanghut@gmail.com (P.); pujosumantoro@gmail.com (P.S.)
2. Research Center for Plant Conservation, Botanic Garden and Forestry, National Research and Innovation Agency of Indonesia, Jl. Ir. H. Juanda No. 13, Bogor 16122, Indonesia; liliana.baskorowati@gmail.com (L.B.); rina.l.hendrati@gmail.com (R.L.H.); mudjisusanto@yahoo.com (M.S.); masshudy64@gmail.com (M.); setiadi2009@yahoo.com (D.S.); iluh_nc@yahoo.com (I.L.G.N.); sg_pudjiono@yahoo.co.id (S.P.); sumardi_184@yahoo.com (S.)
3. Research Center for Ecology and Etnobiology, National Research and Innovation Agency of Indonesia, Jl. Raya Jakarta Bogor Km 45, Bogor 16911, Indonesia; age_kurniawan@yahoo.com
4. Faculty of Forestry, Universitas Gadjah Mada, Yogyakarta 55182, Indonesia
* Correspondence: pandu.yudha.a.p@ugm.ac.id; Tel.: +62-274-548815

**Abstract:** The native tropical pine (*Pinus merkusii* Jungh. and de Vriese) has been genetically improved in Indonesia since 1977; nevertheless, minor evaluations of aphid resistance have been conducted since 2004. As a result, a progeny test for aphid resistance was established in 2010 in Lawu, Central Java, Indonesia. Subjects in the trial were attacked significantly at the rate of 30.7% after 4 years, but surprisingly, some individuals were found to be healthy without any aphid attack. The observed a 7-year progeny trial comprised 34 families with 4 trees per unitary plot and replicated in 10 blocks. At 7 years, observations during 9 months (April–December) showed that there were differences in the range of resistance across families. The stem diameter, oleoresin production, and resistance to aphid attack were evaluated, and all traits were distinct among families except for oleoresin exudation from the western side of the stem. Five families performed above average for all three traits, while three other families had high diameter and maintained good oleoresin production. These eight families can be included in a forward selection strategy. Cluster analysis revealed that the eight best families were grouped into two of the eight clusters. Phenotypic correlations revealed that all pairs of traits were significantly related, with the highest correlation registered between stem diameter and resistance to aphid attack (0.99). Forward selection ensures the simultaneous improvement of the three traits.

**Keywords:** breeding strategy; phenotypic correlation; *Pineus boerneri*; progeny; tropical pine

## 1. Introduction

As a pioneer species, *Pinus merkusii* Jungh. and de Vriese is the only one native pine species in Indonesia, called tusam, which is also found in several Southeast Asian countries [1]. This pine is grown as the second largest artificially regenerated forest after teak in Java for oleoresin production. *Pinus merkusii* has been genetically improved in Indonesia since 1977 [2]. Genetically improved traits include greater stemwood growth, reduced fox tailing, and increased oleoresin production [3–6]. There were minor evaluations of aphid resistance in this species, with the well-known aphid (*Pineus boerneri* Annand, 1928) having been recorded as damaging *P. merkusii* and other pine species globally, including *P. kesiya* Royle ex Gordon [7], *P. tecunumanii* Eguiluz and Perry, *P. maximinoi* H.E. Moore, *P. oocarpa* Schiede ex Schltdl [8], *P. taeda* L., *P. elliottii* Engelmann, and *P. caribaea* Morelet [7].

Aphid species *P. boerneri* is extensively spreading in *P. merkusii* stands, causing significant economic loss in numerous places across Java, Indonesia since 2004 [9–11]. Furthermore, this species attacks the same pine species in Kalimantan and the Maluku Islands, Indonesia [12,13]. This pest brings about foliage discoloration, biomass reduction, branch distortion, decreases in seed quality and quantity, and a decline in oleoresin production. In some productive stands, the effects of aphid infestation have caused tree death [10,14]. Outbreaks of this pest can be controlled by aerial fogging and stem infusions of liquid containing pesticides. However, these options only have temporary effects due to the earlier establishment of broods that cause reinfestation. Extensive and repeated fogging is expensive and unsuccessful in identifying the resistant genotype of *P. merkusii* to *P. boernierii* in order to ensure the maintenance of oleoresin production [15]. *Pinus boerneri* has globally infested more than 40 pine species, ut might damage or kill trees [16], and has the potential to become a serious threat for *P. merkusii*. The relatively long rotation age of *P. merkusii* requires a solution for eradicating this aphid pest to conserve the energy, time, and cost of *P. merkusii* plantation establishment and management [17].

A progeny test for aphid resistance was established in 2010 in Lawu, Central Java with genetic material originating from individuals with the best stemwood growth in a first-generation seed orchard. At the age of 4 years, the trial subjects were significantly attacked at an average rate of 30.7% of the total individuals. At the same time, some individuals were surprisingly healthy without any aphid attack. These observations indicated that, in the 2010 progeny trial, there was a difference in *P. boerneri* resistance across families. This variation enabled the selection of resistant genotypes for more robust future *P. merkusii* plantations. In addition to aphid resistance, stem diameter and oleoresin production were measured because these three traits represent the economic potential of *P. merkusii* in Indonesia.

In the future, healthier and more productive genetic material for large-scale *P. merkusii* plantations could contribute to increasing the value of *P. merkiusii* for society. The purpose of this study was to examine the genetic potential of 34 families for stem diameter growth, oleoresin production, and aphid resistance, and to discover possible relationships that exist among those traits in a field trial. This effort would help in sustaining the production of oleoresin in Java, which achieves up to 60,000 tones year$^{-1}$ from around 800,000 ha of plantation [18].

## 2. Materials and Methods

### 2.1. Trial Site

The experimental site was located in the village of Mendak, in the Ponorogo district of Eastern Java, Indonesia (111°42′7.67″ S, 7°44′6.64″ E; Figure 1). The topography is hilly with slopes between 5% and 15% and an altitude of 869 m above sea level. An average daily temperature of 25 °C with a minimum of 18 °C and a maximal daily temperature of 31 °C were recorded in 2010. In the 2016–2020 period, the sum of the annual precipitation ranged between 2500 and 2958 mm year$^{-1}$. The soil type at the trial site is andosol.

### 2.2. Experimental Design of Progeny Test

The second-generation seed orchard (SO) used in this study comprised the 34 best families from a first-generation SO. The second-generation SO was designed in a randomized complete block design (RCBD) with 4 trees per unitary plot and per block, with 10 blocks (replications). Individuals were planted at 4 m × 4 m spacing.

### 2.3. Data Collection

At the start of the 9-month monitoring period, no mortality due to aphid attack had occurred among the 34 families, although a lack of growth survival initially occurred on some individuals. The evaluation process was conducted in 2010 for 9 months from April to December. Three traits were selected for evaluating this experiment, namely, diameter at breast height (DBH), oleoresin production, and plant resistance to aphid attack (PR).

Diameter at breast height for each tree was measured at 1.3 m from the ground by using a diameter tape. Oleoresin production was observed on the eastern and western sides of the stem by drilling (diameter 10 mm) into the main stem with a drilling position of 50 cm above groundline. After 3 days, oleoresin was collected and assessed by fresh weight [19].

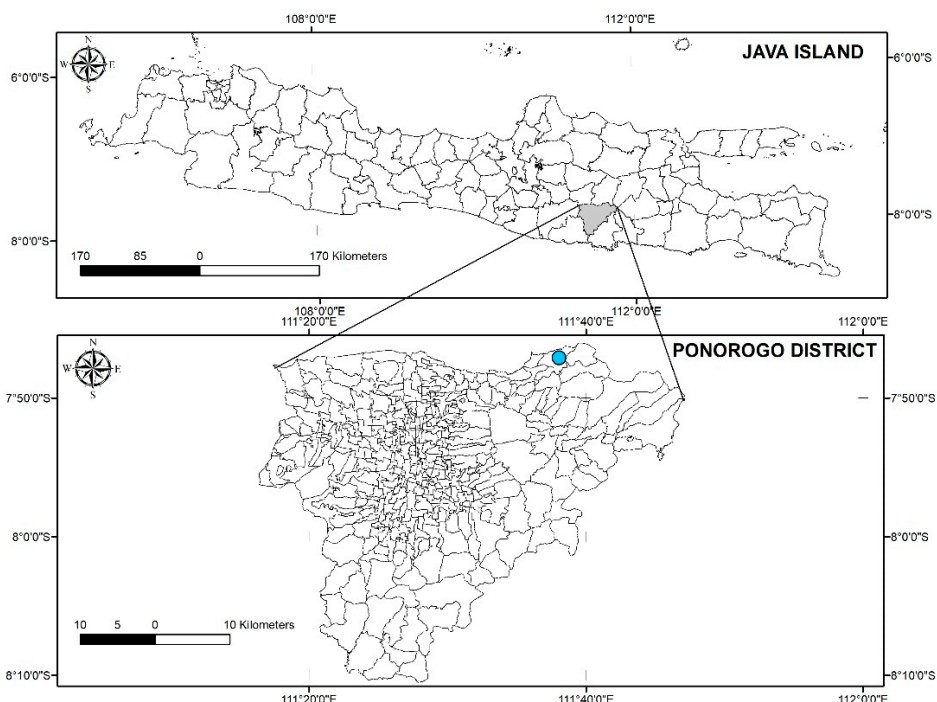

**Figure 1.** Site location of *P. merkusii* second-generation seed orchard, indicated by blue circle.

Resistance to aphid infestation was measured by crown appearance. Crown observations were implemented with a scoring system as follows: 1 = severe aphid infestation with some of the top crown dying back; 2 = heavy aphid infestation (the most susceptible) with >50% of the crown having whitish points; 3 = medium aphid infestation with 25–50% of the crown having whitish points; 4 = low aphid infestation with <25 of the crown having whitish points; and 5 = healthy crown having no whitish points (resistant).

*2.4. Data Analyses*

The means of the 4-tree plots were calculated and applied in statistical analysis, and processed using R software (Version 4.1.1, Vienna, Austria) with a 5% level of significance [20]. If the data did not follow normality distribution and homogeneous variance, natural logarithmic transformation was applied for data transformation. Before analyses, the normality of plot-level data was examined with the Shapiro–Wilk test [21], and the homogeneity of variance among plot-level data was evaluated with Bartlett's test [22]. The comparison of family performance was analyzed using ANOVA with the linear effects model:

$$Y_{ijk} = \mu + B_i + F_j + B_i(F)_j + \varepsilon_{ijk},$$

where $Y_{ijk}$ is the plot mean of the *j*-th family in the *i*-th block; $\mu$ is the overall mean; $B_i$ is the effect of the *i*-th block; $F_j$ is the effect of the *j*-th family; $B_i(F)_j$ is the interaction effect of the *i*-th block and *jk* family; and $\varepsilon_{ijk}$ is the residual error. Mean separations were conducted with Duncan's multiple-range test. Subsequently, hierarchal-cluster analysis was conducted to categorize the families into groups on the basis of their levels of stem diameter growth, oleoresin production, and resistance to aphid attacks [23]. Relationships among these three traits were assessed with Pearson's correlation [24].

## 3. Results

Results demonstrate that the all traits were significantly affected by the blocks. Each block differed, and blocks had a strong relationship with all traits (Table 1). There were block and family interactions in all traits. Similarly, all traits were distinct among families except for oleoresin production recorded from the west. By utilizing the T-test, oleoresin yield from the east showed no significant difference from that produced in the west. Additionally, the diameter, which measures growth due to its possibility to influence oleoresin production, revealed the highest probability of significance.

**Table 1.** Analyses of variance of *P. merkusii* diameter (cm), eastern, western, and average oleoresin production (g tree$^{-1}$), and plant resistance to aphids at Year 7 of a 7-year-old progeny trial.

| Source of Variation | *df* | Sum of Squares | Mean Square | F | *p*-Value |
|---|---|---|---|---|---|
| Diameter at breast height | | | | | |
| Block | 9 | 186 | 20.67 | 4.88 | <0.001 ** |
| Family | 33 | 105.74 | 3.20 | 1.50 | 0.048 * |
| Block × family | 217 | 880.24 | 4.06 | 1.90 | <0.001 ** |
| Error | 213 | 455.78 | 2.14 | | |
| West oleoresin production | | | | | |
| Block | 9 | 1678.90 | 186.55 | 8.12 | <0.001 ** |
| Family | 33 | 957.60 | 29.02 | 1.26 | 0.170 ns |
| Block × family | 217 | 6887.10 | 31.74 | 1.38 | 0.003 * |
| Error | 213 | 4018.30 | 22.96 | | |
| East oleoresin production | | | | | |
| Block | 9 | 958.9 | 106.55 | 5.39 | <0.001 ** |
| Family | 33 | 1098.1 | 33.28 | 1.68 | 0.017 * |
| Block × family | 217 | 4477.8 | 20.64 | 1.04 | 0.217 ns |
| Error | 213 | 3459 | 19.77 | | |
| Average oleoresin production | | | | | |
| Block | 9 | 1646.20 | 182.91 | 11.47 | <0.001 ** |
| Family | 33 | 971.70 | 29.44 | 1.85 | 0.005 ns |
| Block × family | 217 | 5077.80 | 23.40 | 1.47 | 0.002 ** |
| Error | 213 | 3397.30 | 15.95 | | |
| Plant resistance to aphids | | | | | |
| Block | 9 | 38.36 | 4.26 | 15.28 | <0.001 ** |
| Family | 33 | 17.86 | 0.54 | 1.93 | 0.046 * |
| Block × family | 217 | 102.88 | 0.47 | 1.69 | <0.001 ** |
| Error | 213 | 56.67 | 0.28 | | |

Note: *df*, degree of freedom; *F*, value of F table; *, significant at the 0.05 level; **, significant at the 0.01 level; ns, nonsignificant.

From the DBH with a range of 15.9–17.6 cm, western oleoresin production of 5.4–14.5 g, eastern oleoresin production of 5.2–11.3 g, average oleoresin production of 4.6–12.9 g, and a resistance value of 3.3–4.1, some families performed above average for all three significant traits, namely, Families 2, 18, and 28; those with high resistance and good oleoresin production were Families 1, 10, 11, 19, and 32 (Table 2).

On the basis of observations through cluster analyses, the optimal number of groupings was eight (Figure 2). This group was constructed on the basis of the similarity performance of families assessed from the four traits of DBH, eastern and western oleoresin production, and plant resistance to aphids. Trait evaluations within each cluster, together with family members of clustering results (Table 3), demonstrate that the eight best performers were included in two clusters. Cluster 1 was the best group, with a family member having three good traits in DBH, eastern oleoresin production, and resistance. Cluster 6 comprises three families with two good traits in DBH and oleoresin production. Therefore, this result complies with the best family performers in the Duncan post hoc analysis presented in Table 2.

**Table 2.** Results of Duncan's test for *P. merkusii* families at Year 7 of a 7-year-old progeny trial.

| Family | DBH (cm) Mean ± sd | WOP (g) Mean ± sd | EOP (g) Mean ± sd | AOP (g) Mean ± sd | PR Score Mean ± sd |
|---|---|---|---|---|---|
| F1 | 17.6 ± 2.0 ab | 7.8 ± 5.4 bc | 9.5 ± 7.6 abc | 7.2 ± 5.3 bcde | 3.8 ± 0.8 abcd |
| F2 | 17.6 ± 1.7 ab | 9.9 ± 6.4 bc | 9.7 ± 3.2 abc | 8.7 ± 3.8 bcd | 3.6 ± 0.8 abcd |
| F3 | 16.7 ± 1.7 ab | 9.0 ± 7.3 bc | 8.3 ± 4.1 abc | 8.6 ± 5.2 bcde | 3.7 ± 1 abcd |
| F4 | 16.9 ± 1.9 ab | 9.4 ± 5.7 bc | 7.2 ± 3.5 abc | 7.5 ± 4 bcde | 3.8 ± 0.4 abcd |
| F5 | 15.9 ± 1.6 b | 9.8 ± 4.3 bc | 6 ± 4.2 bc | 6.3 ± 3.9 cde | 3.8 ± 0.5 abcd |
| F6 | 16.9 ± 2.1 ab | 7.5 ± 5.1 bc | 5.9 ± 5.4 bc | 6.1 ± 4.9 cde | 3.4 ± 0.6 cd |
| F7 | 16.8 ± 1.8 ab | 5.4 ± 4.3 c | 5.6 ± 3.7 c | 5.0 ± 3.6 de | 3.7 ± 0.7 abcd |
| F8 | 17.3 ± 2.2 ab | 7.7 ± 7.4 bc | 5.7 ± 2.9 c | 6.1 ± 4.6 cde | 3.6 ± 0.6 bcd |
| F9 | 16.8 ± 1.7 ab | 6.0 ± 3.6 c | 6.1 ± 3.1 bc | 5.7 ± 2.8 cde | 3.9 ± 0.6 abc |
| F10 | 18.0 ± 2.0 a | 9.3 ± 7.5 bc | 7.6 ± 4 abc | 8.0 ± 5.0 bcde | 3.8 ± 0.8 abcd |
| F11 | 17.2 ± 1.3 ab | 9.3 ± 6.5 bc | 10.7 ± 5.9 ab | 8.6 ± 4.9 bcde | 4.1 ± 0.8 a |
| F12 | 15.9 ± 0.5 ab | 14.5 ± 6.0 a | 11.3 ± 3.7 a | 12.9 ± 3.3 a | 4 ± 0.8 ab |
| F13 | 16.2 ± 2.3 ab | 8.7 ± 5.0 bc | 7.4 ± 4.3 abc | 8.0 ± 4.1 bcde | 3.5 ± 0.7 bcd |
| F14 | 17.0 ± 2.1 ab | 7.0 ± 3.2 bc | 6.9 ± 4.9 abc | 6.9 ± 3.3 cde | 3.9 ± 0.6 abcd |
| F15 | 17.2 ± 2.3 ab | 9.7 ± 7.1 bc | 5.9 ± 2.7 bc | 7.8 ± 4.5 bcde | 3.6 ± 0.7 bcd |
| F16 | 17.2 ± 2.3 ab | 6.8 ± 3.6 bc | 5.4 ± 3.5 c | 6.1 ± 3.3 cde | 3.6 ± 0.7 bcd |
| F17 | 16.9 ± 1.4 ab | 8.8 ± 9.2 bc | 9.2 ± 6.7 abc | 7.5 ± 7.3 bcde | 3.5 ± 0.5 bcd |
| F18 | 16.9 ± 2.1 ab | 9.1 ± 6.3 bc | 10 ± 7.4 abc | 9.3 ± 6.4 bc | 3.6 ± 0.7 abcd |
| F19 | 16.5 ± 1.5 ab | 6.8 ± 4.3 bc | 8.8 ± 7.5 abc | 7.9 ± 5.6 bcde | 3.5 ± 0.7 bcd |
| F20 | 16.5 ± 1.6 ab | 7.9 ± 5.4 bc | 6 ± 2.7 bc | 6.7 ± 4.1 cde | 3.5 ± 1 bcd |
| F21 | 16.0 ± 2.7 ab | 7.3 ± 5.5 bc | 6.3 ± 3 bc | 4.5 ± 3.9 e | 3.8 ± 0.9 abcd |
| F22 | 17.3 ± 2.3 ab | 6.3 ± 4.5 bc | 5.4 ± 5.7 c | 4.7 ± 4.3 de | 3.7 ± 0.8 abcd |
| F23 | 16.9 ± 1.8 ab | 6.2 ± 4.9 bc | 7.1 ± 3 abc | 5.9 ± 3.6 cde | 3.4 ± 0.5 cd |
| F24 | 17.2 ± 1.9 ab | 8.6 ± 4.9 bc | 7 ± 2.4 abc | 7.4 ± 3.5 bcde | 3.6 ± 0.8 bcd |
| F25 | 17.2 ± 1.8 ab | 8.1 ± 7.0 bc | 6.1 ± 3.5 bc | 6.7 ± 5.1 cde | 3.8 ± 0.4 abcd |
| F26 | 17.3 ± 2.3 ab | 7.6 ± 5 bc | 5.6 ± 2 c | 6.6 ± 3.3 cde | 3.5 ± 0.5 bcd |
| F27 | 16.3 ± 1.7 ab | 6.6 ± 4.9 bc | 7.8 ± 3.9 abc | 6.1 ± 4.3 cde | 3.6 ± 0.7 bcd |
| F28 | 17.6 ± 2.6 ab | 10.6 ± 8.2 abc | 9.3 ± 6 abc | 9.5 ± 6.7 abc | 3.5 ± 0.9 bcd |
| F29 | 16.1 ± 1.4 ab | 8.3 ± 5.0 bc | 5.3 ± 2.9 c | 6.5 ± 3.4 cde | 3.5 ± 0.5 bcd |
| F30 | 17.0 ± 1.8 ab | 5.5 ± 2.7 c | 5.2 ± 3.4 c | 4.6 ± 2.9 e | 3.8 ± 0.6 abcd |
| F31 | 15.9 ± 1.1 b | 11.4 ± 9.9 ab | 10.6 ± 7.9 ab | 11 ± 7.7 ab | 3.7 ± 0.8 abcd |
| F32 | 16.8 ± 2.6 ab | 7.4 ± 4.4 bc | 9.3 ± 4.1 abc | 7.9 ± 3.3 bcde | 3.6 ± 0.6 bcd |
| F33 | 16.9 ± 1.7 ab | 8 ± 4.6 bc | 8.5 ± 6.3 abc | 6.8 ± 5.3 cde | 3.5 ± 0.7 bcd |
| F34 | 16.6 ± 1.4 ab | 8.4 ± 5.5 bc | 8.3 ± 5.8 abc | 8.8 ± 5.4 bcd | 3.3 ± 0.5 d |

Note: DBH: diameter at breast height, WOP: western oleoresin production; EOP: eastern oleoresin production; AOP: average oleoresin production; PR: plant resistance. The mean followed by the same lowercase letter was not significantly different with Duncan's multiple-range test at an alpha level of 0.05.

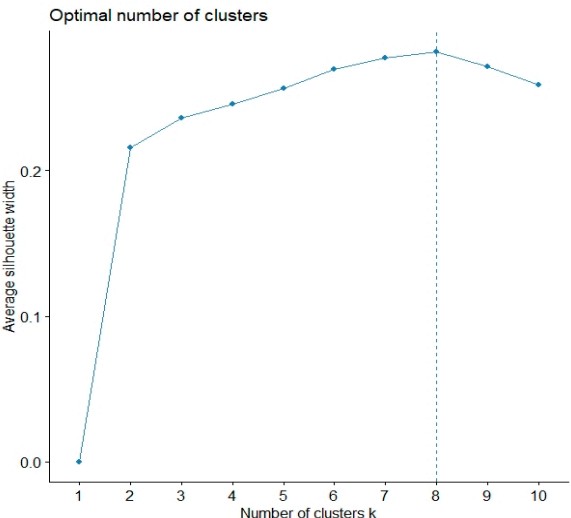

**Figure 2.** Optimal number of clusters of *P. merkusii* families based on the mean of four observed traits.

**Table 3.** Results of cluster analysis of 34 *P. merkusii* families in a 7-year-old progeny trial.

| Cluster | Mean ± sd of Trait Evaluation | | | | Family Membership |
|---|---|---|---|---|---|
| | **DBH (cm)** | **WOP (g)** | **EOP (g)** | **PR score** | |
| C1 | 14.52 ± 0.68 b | 7.42 ± 0.90 cd | 8.32 ± 0.75 b | 3.68 ± 0.08 a | f11, f19, f10, f1, f32 |
| C2 | 12.45 ± 0.78 c | 11.6 ± 0.28 a | 10.3 ± 0.42 a | 3.40 ± 0.00 cd | f12, f31 |
| C3 | 14.30 ± 0.56 b | 8.37 ± 0.79 bc | 6.93 ± 1.00 c | 3.44 ± 0.09 bc | f20, f26, f24. f4, f17, f13, f34, f15, f3 |
| C4 | 12.45 ± 0.49 c | 7.40 ± 0.57 cd | 6.45 ± 0.64 cd | 3.60 ± 0.14 a | f14, f5 |
| C5 | 14.06 ± 0.38 b | 6.50 ± 1.23 de | 5.5 ± 0.49 de | 3.70 ± 0.07 a | f16, f25, f29, f21, f9 |
| C6 | 15.57 ± 0.31 a | 9.20 ± 0.95 b | 9.13 ± 0.68 ab | 3.57 ± 0.06 a | f28, f18, f2 |
| C7 | 15.50 ± 0.55 a | 5.66 ± 1.15 e | 5.00 ± 0.51 e | 3.64 ± 0.09 a | f22, f30, f7, f6, f8 |
| C8 | 14.37 ± 0.40 b | 5.70 ± 0.17 e | 6.83 ± 0.91 c | 3.30 ± 0.00 d | f23, f27, f33 |
| *p*-value | <0.001 ** | <0.001 ** | <0.001 ** | <0.001 ** | |

Note: DBH: diameter at breast height; WOP: western oleoresin production; EOP: eastern oleoresin production; AOP: average oleoresin production; PR: plant resistance. Mean followed by the same lowercase letter was not significantly different by Duncan's multiple-range test at an alpha level of 0.05. ** = significant at level of 0.01

Phenotypic correlations (Table 4) revealed that all pairs of traits are significantly related, with the highest being between DBH and resistance (0.99).

**Table 4.** Phenotypic correlation between traits of *Pinus merkusii* in a 7-year-old progeny trial.

| Trait | Phenotypic Correlation | | | | |
|---|---|---|---|---|---|
| | **DBH** | **WOP** | **EOP** | **AOP** | **PR** |
| DBH | | - | - | | - |
| WOP | 0.938 ** | | - | | - |
| EOP | 0.935 ** | 0.932 ** | | | - |
| AOP | 0.953 ** | 0.964 ** | 0.981 ** | | |
| PR | 0.992 ** | 0.926 ** | 0.927 ** | 0.942 ** | |

Note: ** = significant at level of 0.01; DBH: diameter at breast height; WOP: western oleoresin production; EOP: eastern oleoresin production; AOP: average oleoresin production; PR: plant resistance.

## 4. Discussion

Significance differences between families were found for DBH, oleoresin production, and resistance to aphid attack, suggesting that family selections lead to enhanced stemwood growth, oleoresin yield, and aphid resistance in *P. merkusii*. In this experiment, the use of stem diameter to represent tree growth instead of tree height was based on cases from other experiments, indicating high correlation between stem diameter and tree height for pine species, especially at this age [25–28]. As a result of this positive relationship, selecting at least one trait is sufficient. Furthermore, a previous study found low correlation ($r = 0.5$) between oleoresin yield and tree height, but moderate correlation ($r = 0.8$) between oleoresin yield and stem diameter, indicating that stem diameter is a better predictor of oleoresin yield compared to tree height [4]. Stem diameter was also considered to be the optimal trait to indirectly screen high-oleoresin-yielding individuals in *P. massoniana* Lamb [27].

The collection of oleoresin from the western and eastern sides of the stem in this study was based on past reports of the influence of sunlight on oleoresin yield, with greater yields from the eastern side of the stem [29,30]. Tapping oleoresin from the east is the common practice by the Perhutani State Forest Company in Indonesia. In the summer, when sunlight is more abundant, oleoresin harvest in pines produces more oleoresin than that in winter [31]. Similarly, for *acacia* species, high temperature also increases oleoresin yield [32,33], with low temperatures bringing about the closure of the gum-secreting

tissues [29]. In *Acacia Senegal* (L.) Willd, oleoresin yield was up to 60% higher when tapping took place on the eastern and western sides of the tree compared to the northern and southern sides of the tree [31]. In this situation, the easter and western aspects took advantage of concentrated sunlight [33]. These researchers found that oleoresin exudation from the eastern side of trees demonstrated a more robust family effect than that of oleoresin exudation from the western side of trees. Furthermore, oleoresin yields from the eastern side of the best families were more than double those from the western side of the poorer families (4.4–10.6 g). Family differences among the eastern measurements suggest that it may be appropriate to assess oleoresin yield from the east to minimize experimental error introduced by the stand environment around the circumference of the stem.

Family variationn in aphid resistance in this study is prospective for health improvement in *P. merkusii*. Earlier records disclosed that the aphid attacks had already occurred. Variation was observed at the time of a previous evaluation when 67 individuals were recorded without any aphid attack at all at 4 years of age [34]. Consequently, with the current result, potential improvement is evident. Precaution when providing healthy genotypes of this tropical pine species is undoubtedly essential because outbreaks of the *P. boerneri* aphid in three species of pines have been recorded in Colombia, believed to be a consequence of tropical environmental conditions. Those severe attacks caused the number of aphid generations to be about 3.7 per year [35]. Therefore, the resistance of healthy individuals found in this study is quite meaningful, and variation in family resistance was manifested. Interaction with blocks that appeared in this study indicated some environmental influence. Consequently, to guarantee resistance, observation is suggested to be repeated at later ages.

In our study, family clustering categorized and separated families by their performance relative to four traits: stem diameter, oleoresin yield on the eastern and western sides of the tree, and resistance to aphid attack. In Cluster 1, five families possessed the greatest values in DBH, oleoresin yield, and resistance to aphid attack, suggesting that these families may be characterized by superior stemwood growth, byproduct yield, and forest health. For the three families of Cluster 6, DBH and oleoresin yield were relatively high. The eight families in Clusters 1 and 6 comprised 23.5% of our experimental families, which promise successful genetic gain. This percentage is even higher than the common percentage (10%) for retaining the best genotypes for good genetic gain [36–38]. These promising families that are projected to retain aphid resistance and good oleoresin yield in succeeding years require further investigation by testing them through the establishment of progeny trials.

Strong positive phenotypic correlations among the analyzed traits were recorded in this study. Our observation of a strong correlation between oleoresin yield and plant resistance to aphids (0.94) confirmed other research findings that state oleoresin production increases resistance to beetle infestation [28,39]. The purpose of establishing *P. merkusii* stands in Java is primarily for oleoresin production to support the turpentine oil industry before mature tree harvest for stemwood biomass. Consequently, the ideal genotypes for plantations are those that resist pests, produce oleoresin, and grow at an acceptable biomass accumulation rate. Results of our study advocate the establishment of *P. merkusii* families from Cluster 1, where landowners desire high rates of stemwood and oleoresin production, and the sustained resistance of trees to aphid infestation.

In regard to tree breeding for pest resistance, due to its complex influence of environmental factors, genetic molecular markers should be developed as tools. In the future, the markers would be useful in differentiating resistant genotypes in breeding programs. Further, the development of consensus genetic maps for oleoresin production and aphid resistance in *P. merkusii* is needed to accelerate the tree selection process in breeding strategies. Similar consensus genetic maps using molecular markers have been developed for the *P. taeda* L. and *P. elliottii* Engelm. oleoresin canal [28], and for *P. flexilis* E. James disease resistance [40]. Genetic markers are neutral and not affected by environmental variables, but they are affected by genes, and pest resistance traits are controlled by both specific

single genes and multiple genes. When the genes that control resistance can be identified, the stable character on individuals can be traced regardless of the environment.

In the natural distribution of *P. merkusii*, this species tends to have a high outcrossing rate, partly due to its light and dry pollen, which is dispersed by the wind. Further, the winged seeds also facilitate distant dispersal [41]. Slope is also a possible constraint to pollen and seed dispersal. These impact *P. merkusii* genetic diversity, which commonly moderates its value. This is in contrast with populations from the mountain of Kerinci, which have low genetic diversity due to hilly conditions that drive high selfing rates [42].

## 5. Conclusions

Variation among the traits of stem diameter, oleoresin yield, and resistance to aphid attack in our progeny trial indicated that the present *P. merkusii* breeding program in Mendak Ponorogo, Indonesia is successful. The best families for all traits were Families 1, 10, 11, 19, and 32, while those having two good traits were Families 2, 18, and 28. Strong positive correlations were recorded between traits. The assessment of oleoresin yield among the families was more robust when measurements were observed on the eastern side of the tree compared to the western side of the tree.

**Author Contributions:** P., L.B., P.S., R.L.H., M.S., M., D.S., I.L.G.N., S.P., A.K., P.Y.A.P.W. and S. have contributed equally towards establishing the trial, data recording, data analysis, and preparing and writing this paper. All authors have read and agreed to the published version of the manuscript.

**Funding:** This research was fully funded by the Department Research and Innovatiom, Perhutani Forest Institute, Cepu, Central Java, Indonesia.

**Institutional Review Board Statement:** Not applicable.

**Informed Consent Statement:** Not applicable.

**Data Availability Statement:** Data sharing is applicable upon request.

**Acknowledgments:** We would like to express our sincere gratitude to Perum Perhutani for providing the trial subjects for evaluation, support, and funding to undertake this experiment. We also wish to acknowledge the team for their dedication, especially Suryanaji from the Department Research and Innovation Perum Perhitani and all technicians at Perum Pehutani Ponorogo Distict who recorded the data and collected the oleoresin samples. We also thank to Eko Pujiono, Reseacher from Research Center for Ecology and Etnobiology, National Research and Innovation Agency of Indonesia who illustrated the study site map.

**Conflicts of Interest:** The authors declare no conflict of interest.

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
