# Peer review of "Evaluation of Aphid Resistance and Oleoresin Production in Indigenous Tropical Pine (Pinus merkusii Jungh. and de Vriese)"

_forests, doi:10.3390/f13070977_

Round 1

Reviewer 1 Report

The manuscript was revised and the authors addressed most of my concerns. But I need a clear explanation regarding Table 1 of the previous paper (i.e., Forests 2022, 13(3), 451; https://doi.org/10.3390/f13030451) versus the Table 1 in this manuscript.  If the paper was submitted as a research article I will ask for a major revision because the results in phenotypic correlation and family-wise study were not typically influential for tree breeding programs without the genetic analysis.  Because the paper was submitted as a short communication, I will ask for a further response to my comment.  Some of the discussion could be very self-referenced and the more important findings for the tree breeding audience were already published except for the family clusters.  The authors can provide an inferential result regarding the comparison of the family groups.  If the family groups are significantly different from each other, this will be another piece of message for the group improvement other than individual tree breeding.  Any historical provenance effect exists here? The author can provide some pictures of the aphid, resistance to the aphid, and oleoresin production in the text for the readers.  

Author Response

Dear Reviewers

Many thanks for your valuable inputs and comments for improving the quality of this manuscript. Please find my responses to your comments in the attach file. 

Best regards 

Reviewer 2 Report

Dear Authors,

I have read the manuscript “Evaluation of Aphid Resistance and Oleoresin Production in Indigenous Tropical Pine (Pinus merkusii Jungh. & de Vriese)” several times, and, in my opinion, the paper can be interesting for the scientific community. However, I have some recommendations, which in my opinion will help the reader to understand better the new information brought by the paper. I have to recommend a minor revision of the manuscript.

General Comment:

The major weaknesses of the paper are related to the existence of doubts related to the correct use of statistical programs (especially the ANOVA test), as well as to the uninspired use of some graphs or tables.

Specific comments:

Abstract

- Line 18: I suggest adding a comma after 2010.

- Line 21: ‘four trees per unitary plot’ instead of ‘4-tree plots’.

- Line 23: try to replace ‘disparities’.

- Line 25: replace ‘in’ with ‘for’.

- Line 26: add ‘other’ after ‘while’. Also, add the sentence ‘These 8 families can be included in a forward selection strategy’ after ‘production’.

- Line 27: replace ‘demonstrated’ with ‘revealed’, delete ‘performers’, replace ‘assembled into’ with ‘grouped in’, and add ‘the’ after ‘two of’.

- Line 29: replace ‘being’ with ‘registered’. In the end, add the sentence ‘Forward selection will ensure the simultaneous improvement of the three traits’.

Keywords

- Line 30: In alphabetical order. Replace ‘families’ with ‘breeding strategy’.

1. Introduction

- Line 33: I suggest you to delete ‘of’ and add ‘one’ after ‘only’.

- Line 36: Add a comma after ‘Indonesia’ and delete ‘late’. Try to use the common name of the species Pinus merkusii at least a few times.

- Lines 52-53: replace ‘ineffective’ with ‘unsuccessful’. ‘..to identify the resistant genotype of P. merkusii to P. boernerii, in order to ensure…’.

- Line 54: replace ‘Pinus’ with ‘P.’ or ‘Pineus’.

- Line 56: replace ‘to’ with ‘for’.

- Line 61: delete ‘seedling’ and start the next sentence with ‘At the age of 4 years’, and a little further change ‘attacked significantly’ with ‘significantly attacked’.

- Line 65: replace ‘better performing’ with ‘resistant’.

- Lines 69-70: delete the sentence.

- Line 72: delete ‘or even’ and replace ‘to society’ with ‘for society’.

- Line 74: add ‘to’ before ‘discover’.

- Line 75: add a comma after ‘traits’ and replace ‘this’ with ‘a field’.

- Line 76: replace ‘ton’ with ‘tones’.

2. Materials and Methods

- Line 80: replace ‘was’ with ‘is located’.

- Lines 83-84: add ‘with’ before ‘a minimum’ and remove the repetition of ‘daily temperature’. Start the next sentence with ‘In the 2006-2010 period, the sum of the annual precipitation ranged between 2500 and 2958…’.

- Line 87: Figure 1, delete ‘seedling’.

- Lines 91-92: at 2.2. subchapter, ‘four trees per unitary plot and per block, with 10 blocks (replications)’.

- Line 95: replace ‘plots of the 10 blocks’ with ‘families’.

- Lines 103-108: code 1 is missing. Only 2 to 5 are described.

- Line 116: replace ‘by analyses of variance’ with ‘using ANOVA,’ in order to avoid the repetition.

- Line 122: remove ‘with a mean of zero’.

3. Results

- Lines 137-138: in Table 1, the first line of the table is missing (DF, SS, MS, F, p). Please explain how these DF values were reached. For me, Block x Family = 9 x 33= 297. The Error DF must be higher. Please explain. Maybe I didn't get it right. It is very important.

- Lines 141-142: in Table 2, the title is inside the table. Why not present as a Duncan test, homogeneous groups (ranking)? It would be much more suggestive.

- Line 148: replace ‘diameter’ with ‘DBH’.

- Line 150: replace ‘in’ with ‘for’.

- Lines 153-154: in Figure 2, the most efficient cluster analysis was not used. The one in which the families also appear could be much more suggestive.

- Line 165: add ‘included’ after ‘were’.

- Line 170: table 4, replace ‘year 7 of a’ with ‘a’.

4. Discussion

- Lines 178-265: Try to reduce this section a bit. E.g.: Too many explanations for using dbh instead of Th, etc.

- Lines 179-180: start with ‘Significance differences between families were found for DBH, oleoresin production and resistance to aphid attack, suggesting that family selection will lead to enhance the…’.

- Line 181: replace ‘in’ with ‘of’.

- Line 184: add a comma before ‘especially’, delete ‘4 to 8 years’ and add ‘this’ before ‘age’.

 - Lines 186-187: close the parenthesis after ‘0.8’ and add a comma after ‘diameter’.

- Line 194: add a comma after ‘Company’.

- Line 196: ‘Acacia’ and with italic.

- Lines 230-232: Replace as follows: ‘Strong positive phenotypic correlations between the analyzed traits were recorded in this study’.

- Line 245: remove the blank space between ‘to’ and ‘its’.

- Line 247: add a comma after ‘future’.

- Line 249: remove the blank space between ‘accelerate’ and ‘tree’.

- Line 254: remove the blank space before ‘When’.

- Line 260: ‘merkusii’ all with italic.

- Lines 263-265: Please delete the last two phrases, after ‘[42]’.

5. Conclusions

- Line 270: replace ‘had three good’ with ‘, for all traits’.

- Line 271: ‘f32, while those having only two…’.

- Line 272: replace ‘among the three’ with ‘between’.

Funding

- Line 281: add a comma after ‘Java’.

References

Lines 291-393: You must respect the Instructions for the authors of Forests journal.

- In the whole list, write in italics the scientific name of the species and family. References: 1-11, 13-14, 16, 25-27, 32, 37-38, 40, 42.

- Translate all references into English. References: 6, 11-12, 18-19, 30, 34, 41.

- Translate from Chinese the journal name of the reference 41.

- ‘Vienna’ at line 342.

- Other typos and blanks in the references 8, 10-11, and 27.

Author Response

Dear reviewers,

Many thanks for your valuable input and comments. Please find our response to the comments at attached file.

Best regards,

Lili

This manuscript is a resubmission of an earlier submission. The following is a list of the peer review reports and author responses from that submission.

Round 1

Reviewer 1 Report

Short Communication 1: Evaluation of Aphid Resistance and Oleoresion Production in Indigenous Tropical Pine (Pinus merkusii Jungh. Et de Vriese)

General comments:

The paper is a short communication that aims to examine the genetic potential of some families for these traits (i.e. aphid resistance, growth (i.e. diameter) and oleoresin production and discover possible relationships that might exist in the progeny trial. The progeny trial was properly designed and assessment of growth, oleoresin traits and aphid resistance were done at suitable age for drawing a good conclusion. In particular, we noted that the new data and information resulted from the research is trait for aphid resistance.

Data collection was done at 6 years old progeny trial that was suitable for the traits of interest assessment. The trial comprised 34 families from which each family has four tree plots replicated into 10 blocks.

As a weakness, the research used only one site that cannot capture the environmental variability  including biodiversity effects, environmental processes for defining the suitable sites for Pinus merkusii in Java. The authors need to justify and discuss from previous recent research or publication on current aphid occurrences in Java or other sites and check/add/ use approaches based on the climate conditions of each site.

We also noticed that a quite significant number of the references (n= 16) are not recently published (within 5 years). The authors need to add more recently published references.

Specific comments:

Line 13:

"indigenous" to be changed to "native"

Line 39-45:
Adding more references as to the pest from other cases even from non-tropical climate.

Many natural plantations are adapting to climate changes. Please add reference about the possible threats to Pinus merkusii now and in the future

Line 46:

What chemicals?

Drilling by injecting?

Line 52-61:
Please add problem statement in this paragraph.

Many natural plantations are adapting to climate changes. Please add reference about the threats to P merkusii now and in the future

Line 65-67:
Please add problem statement in this paragraph

The regions >>> forest division?

FMU Lawu as a standard term

Line 86-87:

Please seperate between measuring and scoring the paratmeters

Line 89:

Detail specs for hypsometer

Line 92-93:

Any standard protocol developed before?

Line 100-103:

No such data are presented in the 3. results section. Please present the data in box plots for all parameters at the beginning of the results presentations to give quick overview of performance with respect to the variability in the trial

Line 105-106:

Add references?

Line 107:

Please add data analysis in the form of box plots for:

- diameter

- oleoresin yield

- aphid resistance

Line 129:
Figure 2 is not necessary to be put here...Please explain more on data collection and analysis (section 2.4)

Line 144:
Please add in Discussions:

  1. The use of genetic markers for evaluating pest resistance
  2. The nature of still having broad genetic variation in the population under investigation
  3. Limitations of the experiments

Line 168:

The lowest production >> per 3 days?

Author Response

Dear Reviewer 1,

We have revised the manuscripts as your advised. However, for the data that illustrated into box plot, we are going to put as supplementary file. Another request regarding the experimental layout also will presented as supplementary file.

We did the revision of the manuscript by hi light it using the green colour and please have a look the checklist of authors response presented below the manuscript.

Best regards and many thanks for your valuable comments 

Liliana Baskorowati

Reviewer 2 Report

General comments

This study evaluates aphid resistance and the relationship between aphid resistance, stemwood growth, and oleoresin production during the seventh year of growth in a P. merkusii progeny trial with 34 families. The progeny trial is well designed with 10 blocks containing 34 4-tree subplots. The results identify two subsets of families with superior resin yield and aphid resistance with one of these groups also having higher rates of stemwood growth compared to other families. The findings appear to be of value where oleoresin yield and the role of oleoresin in pest resistance are important.  However, the findings warrant consideration for publication only after very major revisions are made to the paper.  That being said, the technical soundness of the progeny trial and need for information about P. merkusii provide a strong justification for the paper to be improved and reworked for a later submission.

The recommended revisions are very substantive and include providing much more detail about the sampling methods and statistical analyses.  For example, the experimental units (i.e., individual trees or 4-tree subplots) are not defined and appear to be individual trees until the degrees of freedom in Table 1 indicate otherwise.  As another example, the authors report resin yield in two cardinal directions and their significance is not consistent.  Since whole tree resin yield is what would be under genetic control, it is suggested that the two values by tree be averaged to reduce experimental error before ANOVA.  Also, the ANOVA reports a Block x Family effect that is often significant rather than reporting this as experimental error.  If this is acceptable, then more information and explanation is needed about what is causing the significant interaction.  At present, there is no explanation for even how the 10 blocks were delineated.  The Methods section requires more detail.  The tables and figure are not self-explanatory and need more information and consistency in the headings and added footnotes.  With regard to the level of detail needed in the Methods section and the need for tables and figures to be self-explanatory, it is suggested that the authors review and follow the format of published scientific papers on the topic of pine genetic resistance to pests.  There are several needs for improved language and scientific format.  For example, specific epithets are often excluded.  Geographic locations are not descriptive enough for those unfamiliar with Indonesia to understand where this was done.  Mid-sentence capitalizations should be corrected.  Only one term should be used consistently for nouns.  Abbreviations should only be used after they have first been defined in parentheses behind the first mention of the noun.  This applies to tables and figures.  Sentences should not begin with an abbreviation.

Please see the attached file for specific comments.

Author Response

Dear Reviewer 2,

Many thanks for your valuable comments and inputs. We did revise the manuscript as your advise. Please find the author response checklist below the manuscript attached. 

Best regards and many thanks for your comments

Liliana Baskorowati

Reviewer 3 Report

Title: Evaluation of Aphid Resistance and Oleoresin Production in Indigenous Tropical Pine (Pinus merkusii Jungh. & de Vriese)

Manuscript Type: Original Article

Comments:

The author studied the aphid resistance and oleoresin traits of a tropical pine species, which is important for the Southeast Asia forests and forestry industry.  This manuscript provides specific insights on utilizing the among family genetic variation for plantation forestry and tree improvement.  This will attract multiple audiences and stakeholders in forestry, policy making,  and R&D in the tropical forested regions.  . 

This manuscript can be improved in the following aspects, 1) some key points are necessary to be addressed such as the potential of BLUP method for estimating the breeding values of the families though ANOVA which was equivalent to BLUE was used here; 2) this paper lacks the discussion of the genetic correlation between the aphid resistance and oleoresin traits which are needed to corroborate the phenotypic correlations because the genetic correlations are more useful for breeding and selection in the future generations.  However, this manuscript was well revised with a nearly acceptable quality, and such an amount of work deserves publication eventually. But minor revision is needed before acceptance.  My  detailed comments were as follows,

Line 302, some significance should be identified here such as the p-value of the correlation coefficients though Table 4 showed the results.  

Line 227, EOP and height was not presented in the table 4. 

Line 284, This approach is different from the common tree improvement program in operation and provides a novel insight into the future breeding program.  Usually, ideal families were chosen from the breeding program for backward or forward selection.  The clusters could form breeding groups for future generations of tree improvement by increasing the gains within and among the groups while keep potential among group population differences or diversity level.

It is promising to have multiple locations for more tests of the families with more individual trees and families tested to validate the trend found in this paper such as the phenotypic correlation in the future studies.  

A reference should be added in the introduction

Imanuddin, R.; Hidayat, A.; Rachmat, H.H.; Turjaman, M.; Pratiwi; Nurfatriani, F.; Indrajaya, Y.; Susilowati, A. Reforestation and Sustainable Management of Pinus merkusii Forest Plantation in Indonesia: A Review. Forests 2020, 11, 1235. https://doi.org/10.3390/f11121235

Round 2

Reviewer 2 Report

This is a nice short-term study that provides good information about P. merkusii family selection regarding three traits of interest in Indonesia. The authors adequately addressed several of my earlier concerns, but failed to resolve a potentially large problem in their statistical analyses. The study results warrant publication, but much work is yet to be done before publication is possible,  Table 1 describes an ANOVAs with degrees of freedom that are incorrect for a RCBD with 34 families and 10 blocks.  In the study design, the 4-tree sub-plots are the experimental units and block x family is the experimental error. Furthermore, it is unclear if the 4-tree subplot means were used for the cluster and correlation analyses which they should have been. Furthermore, the authors indicate that tests of normality and homogeneity of variance were done. But, there is no information about how they dealt with non-normal traits or if there were any.  The Discussion and Conclusions need clarity.  However, this can only be accomplished after the statistical analyses have been corrected.  Comments to streamline the text have been suggested in the attached file.

Author Response

Dear Reviewer,

On behalf of the authors, we would like to say many thanks for your valuable comments and suggestion to improve this manuscript. We already revised all based your comments and suggestions. 

Many thanks 
